# The Intricate Interplay between the ZNF217 Oncogene and Epigenetic Processes Shapes Tumor Progression

**DOI:** 10.3390/cancers14246043

**Published:** 2022-12-08

**Authors:** Pia Fahmé, Farah Ramadan, Diep Tien Le, Kieu-Oanh Nguyen Thi, Sandra E. Ghayad, Nader Hussein, Chantal Diaz, Martine Croset, Philippe Clézardin, Pascale A. Cohen

**Affiliations:** 1Université Lyon 1, Lyon, France; 2INSERM, Research Unit UMR_S1033, LyOS, Faculty of Medicine Lyon-Est, 69372 Lyon, France; 3Department of Biology, Faculty of Science II, Lebanese University, Beirut, Lebanon; 4Laboratory of Cancer Biology and Molecular Immunology, Department of Chemistry and Biochemistry, Faculty of Science I, Lebanese University, Hadat, Lebanon; 5University of Science and Technology of Hanoi, Vietnam Academy of Science and Technology, Hanoi 11355, Vietnam; 6Center for CardioVascular and Nutrition Research (C2VN), INSERM 1263, INRAE 1260, Aix-Marseille University, 13385 Marseille, France

**Keywords:** ZNF217, oncogene, epigenetics, microRNA, long noncoding RNA, circular RNA, DNA methylation, m^6^A deposition

## Abstract

**Simple Summary:**

The ZNF217 oncogene promotes progression, pluripotency and resistance to therapy in different types of cancer and is associated with increased risk of metastasis. High ZNF217 expression levels are a biomarker of poor prognosis in several cancers. Therapies targeting ZNF217 could be promising in cancer treatment; however, the lack of identification of the full 3D structure of ZNF217 incites investigation of mechanisms of regulation of ZNF217 to delineate alternative candidate therapeutic strategies. In fact, studies show that ZNF217 shapes cancer growth through epigenetic mechanisms that promote tumor progression, including DNA methylation, interactions with noncoding RNAs and epitranscriptome refining, which are summarized in this review. We also discuss how it would be promising to assess noncoding RNAs that regulate ZNF217 expression and function, as well as to investigate the DNA methylation status of *ZNF217* locus, as biomarkers of poor prognosis and to monitor therapeutic response to treatment.

**Abstract:**

The oncogenic transcription factor ZNF217 orchestrates several molecular signaling networks to reprogram integrated circuits governing hallmark capabilities within cancer cells. High levels of ZNF217 expression provide advantages to a specific subset of cancer cells to reprogram tumor progression, drug resistance and cancer cell plasticity. ZNF217 expression level, thus, provides a powerful biomarker of poor prognosis and a predictive biomarker for anticancer therapies. Cancer epigenetic mechanisms are well known to support the acquisition of hallmark characteristics during oncogenesis. However, the complex interactions between ZNF217 and epigenetic processes have been poorly appreciated. Deregulated DNA methylation status at *ZNF217* locus or an intricate cross-talk between ZNF217 and noncoding RNA networks could explain aberrant *ZNF217* expression levels in a cancer cell context. On the other hand, the ZNF217 protein controls gene expression signatures and molecular signaling for tumor progression by tuning DNA methylation status at key promoters by interfering with noncoding RNAs or by refining the epitranscriptome. Altogether, this review focuses on the recent advances in the understanding of ZNF217 collaboration with epigenetics processes to orchestrate oncogenesis. We also discuss the exciting burgeoning translational medicine and candidate therapeutic strategies emerging from those recent findings connecting ZNF217 to epigenetic deregulation in cancer.

## 1. Introduction

The zinc-finger protein 217 (ZNF217) is an oncogenic transcription factor (TF) that plays a key role in tumorigenesis, orchestrating tumor progression in several human cancers at both early and late stages [1]. Evidence suggests that ZNF217 fine tunes a variety of molecular signaling pathways (Figure 1) to reprogram integrated circuits governing hallmark capabilities within cancer cells. ZNF217-driven functions impact several traits of cancer cells involving sustained proliferative signals, evasion from growth suppressors, enabled replicative immortality, resistance to apoptosis, cancer stem cell enrichment, drug resistance and activation of invasion and metastasis (for review [1,2]).

ZNF217 belongs to the Kruppel-like TF family, contains eight predicted C2H2 zinc finger motifs [3] and binds to specific DNA sequences to directly control target gene expression [4,5,6]. ZNF217 was first reported to belong to a transcriptional repressor complex [7], but this oncogene also exerts its deleterious functions through the positive transcriptional regulation of specific gene expression programs [5,8,9], revealing it as a double-sided TF. To date, nothing is known about the existence of any post-translational modification targeting the ZNF217 protein, and aberrant expression levels, thus, represent the main mechanism associated with ZNF217 oncogenic function.

Although few ZNF217 direct target genes have been formally validated, those genes encode for key masters of tumor progression and cancer cell plasticity. ZNF217 directly binds to the promoter and silences the *P15INK4B/CDKN2B* tumor suppressor gene, which encodes for a cyclin-dependent kinase (CDK) inhibitor, thus contributing to ZNF217 oncogenic properties [8]. ZNF217 stimulates cancer cell migration, invasion and epithelial–mesenchymal transition (EMT) by negative regulation of *CDH1/E-cadherin* expression [4,9,10]. In breast cancer (BCa) cells, ZNF217 directly upregulates *HER3* gene expression, facilitating the formation of the well-known HER2/HER3 oncogenic cassette, which results in the activation of the mitogen-activated protein kinases (MAPK) and phosphoinositide 3-kinase (PI3K)/AKT survival pathways [11]. ZNF217 binds to the promoter and positively regulates *TGFB2* and *TGFB3* genes, leading to sustained and autocrine activation of the TGF-β pathway, contributing to ZNF217-driven EMT [9]. In mouse embryonic stem cells, Zfp217, the murine homolog of ZNF217, regulates pluripotency and somatic cell reprograming through the direct positive transcriptional regulation of the pluripotency *NANOG*, *SOX2* and *POU5F1* genes [12]. Finally, the fat mass and obesity-associated *FTO* gene, coding an alpha-ketoglutarate-dependent dioxygenase and mRNA (2′-O-Methyladenosine-N(6)-)-demethylase involved in adipogenesis and epitranscriptome regulation, has been recently demonstrated to be a direct target gene positively regulated by Zfp217 [13].

Aberrant high expression levels of *ZNF217* have been associated with poor prognosis in several human cancers (for review [1,2]). The *ZNF217* locus is located within the 20q13 region, a region frequently amplified in human cancers [3], and amplification at *ZNF217* locus has been associated with poor prognosis in some reports [7,14,15,16]. Although high levels of *ZNF217* expression and amplified *ZNF217* locus may both exist in cell lines and tumors [3,17,18], absence of correlation between *ZNF217* amplification and *ZNF217* expression levels is also reported [3,17,18,19,20]. These observations pinpoint that, aside from *ZNF217* genomic amplification, other molecular mechanisms govern *ZNF217* expression levels. Little is known regarding TFs being formally demonstrated to be recruited at *ZNF217/Zfp217* promoter and to positively regulate the transcription of this oncogene. Recent studies, however, reported that the MYC-associated zinc finger protein (MAZ) and the interferon regulatory factor 5 (IRF5) are direct activators of *Zfp217* transcription, contributing to, respectively, prostate cancer (PCa) and pancreatic cancer aggressiveness [21,22]. In mouse embryonic stem cells, the signal transducer and activator of transcription 3 (STAT3) binds to *ZNF217* promoter to upregulate *ZNF217* expression [23]. In glioblastoma [24] and BCa [25], the hypoxia-induced HIF1α and HIF2α TFs contributed to the maintenance of stem cells by upregulating *ZNF217* expression through a yet undiscovered mechanism. Thus, elucidating the alternative cellular mechanisms contributing to high ZNF217/Zfp217 expression levels and oncogenic functions is of utmost importance.

Epigenetics involves mechanisms that affect the outcome of the genetic code, without altering DNA sequences. These epigenetic processes include DNA methylation, histone modification, chromatin remodeling, alterations in noncoding RNAs (ncRNAs) profile, RNA silencing and deregulation in the epitranscriptome. Change in epigenetic states is closely associated with human diseases, most notably, cancer. Cancer epigenetic processes support the acquisition of hallmark features during tumor progression by contributing to cancer initiation and progression [26,27]. Unraveling epigenetic variations associated with cancer onset, progression and metastasis development is, thus, crucial to improve and delineate candidate strategies for cancer prevention and treatment.

Historically, the first reports describing a collaboration between ZNF217 and epigenetic events are related to chromatin remodeling complexes, involving ZNF217 together with several cofactors and histone-modifying enzymes. ZNF217 was shown to interact with a component of a human histone deacetylase complex (CoREST-HDAC) [4,8,28]. ZNF217 is also found in complexes with the lysine demethylase 1A-LSD1 (H3K4 and H3K9 demethylase), KDM5B/JARID1B/PLU-1 (H3K4 trimethyl demethylase), G9a (H3K9 methylase), EZH2 (H3K27 methylase) and the transcriptional co-repressor C-terminal binding proteins (CtBP) [4,5,7,8,28,29,30,31]. Therefore, ZNF217 is a double-faceted TF with a dynamic property, acting as a scaffold between histone modifiers that assemble into chromatin remodeling complexes, co-ordinating multiple ZNF217-driven oncogenic programs.

Two main previous reviews have extensively depicted the role of ZNF217 in carcinogenesis by affecting the hallmarks of cancers (for review [1,2]), emphasizing ZNF217 importance as a prognostic biomarker for early prevention and as a therapeutic target. Recent literature provides, however, evidence of complex networks involving ZNF217 and epigenetic processes as new key regulators of tumorigenesis. The increasing body of research indicates that epigenetics processes control *ZNF217/Zfp217* expression levels and ZNF217/Zfp217-driven functions. Indeed, deregulated DNA methylation status at *ZNF217* locus or complex cross-talk between *ZNF217* and noncoding RNA networks could explain aberrant *ZNF217* upregulation. On the other hand, ZNF217 could control gene expression signatures and molecular signaling for tumor progression and cell plasticity. ZNF217 has been shown to tune DNA methylation status at key gene promoters to interfere with noncoding RNA or to refine the epitranscriptome.

Altogether, with this topic having been poorly reviewed and appreciated, a better understanding and clear overview of the intricate interplay between ZNF217 and epigenetics networks in cancer is of utmost importance. The present review focuses on and discusses recent advances delineating the complex interaction between ZNF217 and epigenetic processes to orchestrate tumor progression.

## 2. DNA Methylation Status at the *ZNF217* Locus Inversely Correlates with *ZNF217* Expression Levels

DNA methylation status is a well-known process catalyzed by DNA methyl transferase (DNMT) enzymes and recycled during active demethylation by the ten-eleven-translocation (TET) family of enzymes. DNA methylation occurs by the addition of a methyl group to DNA mainly at the 5-carbon position of cytosine (5mC) and almost exclusively in the context of a CpG dinucleotide. The DNA methylation machinery includes writers, editors and readers, which modulate transcriptional machinery [32]. Inappropriate methylation of promoters, control regions, enhancers or insulators governing gene expression is the most relevant DNA methylation change observed in cancer cells. DNA methylation patterns can distinguish cancer cells from normal cells [33,34]. Hypermethylation could result in the silencing of tumor suppressor genes, whereas global DNA hypomethylation observed in cancer leads to genome instability or oncogene activation [35].

Deregulated DNA methylation status at *ZNF217* gene was first highlighted in BCa. DNA methylation status at the *ZNF217* locus was lower in estrogen receptor α-positive (ER+) when compared with ER-negative (ER-) breast tumors [36], while ER+ and ER- breast tumors display, respectively, high and low *ZNF217* mRNA levels [37]. In ER+ BCa cells, ERα signaling disruption by a siRNA strategy triggered epigenetic silencing (hypermethylation) of the *ZNF217* gene [36]. More recently, ectopic overexpression in BCa cells of the exon 4-skipping isoform of ZNF217 (ZNF217-ΔE4) was shown to promote high *ZNF217* wild-type expression levels that were inversely correlated with DNA methylation of the *ZNF217* gene at key CpG sites [38].

In a case–control study involving 1083 blood samples (healthy women *versus* BCa patients), lack of methylation at the *ZNF217* locus in peripheral blood cells predicted BCa risk [39]. More specifically, the DNA methylation status at the *ZNF217* locus predicted invasive ductal but not invasive lobular BCa. A lack of DNA methylation at *ZNF217* locus in peripheral blood cells was associated with high ERα bioactivity in the corresponding serum and was proposed as a surrogate biomarker of estrogen exposure and of BCa risk [39,40].

Using 33 TCGA cancers, a study aiming at clarifying gene expression profile and methylation level in *ZNF* family genes identified that methylation level of *ZNF217* was significantly different between tumor and normal samples, with globally a hypomethylation in tumor samples also found to be negatively correlated with *ZNF217* expression levels [41]. In glioblastoma [42], uveal melanoma [43] and soft tissue sarcoma [44], again, an inverse correlation between *ZNF217* expression levels and DNA methylation status was shown to exist. In glioblastoma *versus* control brain, hypomethylation at the *ZNF217* promoter was paired with increased *ZNF217* expression levels [42]. In a study aiming at identifying DNA methylation profiles in soft tissue sarcoma subtypes, *ZNF217* locus showed DNA hypomethylation and was part of a minimal set of eight CpG sites, allowing the discrimination of the different sarcoma subtypes [44].

Altogether, while compiling evidence highlighted that DNA methylation status is a key event in the transcriptional regulation of *ZNF217* expression levels, unraveling the upstream molecular events that, in a tumor-specific context, might lead to DNA methylation changes at the *ZNF217* locus remains to be elucidated.

## 3. ZNF217 Fine Tunes DNA Methylation Status of the Tumor Suppressor *P15INK4B/CDKN2B* Gene

The *P15INK4B/CDKN2B* gene encodes for a tumor suppressor and is found within the *INK4* locus, which also contains the *p16 ^ink4a^* and *ARF* genes. The encoded p15ink4b/CDN2B protein is a cell cycle inhibitor that acts by blocking the activity of CDK4 and CDK6 [45]. Oncogenic and interrelated events leading to *P15INK4B* silencing, involving repressive transcriptional complexes or DNA hypermethylation are frequently observed in many cancers [46]. ChIP and ChiPseq experiments identified the *P15INK4B* promoter as a direct target of the ZNF217/CoREST complex [8]. Silencing *ZNF217* was associated with increased p15ink4b/CDN2B protein levels, suggesting that *ZNF217* extinction alone was sufficient to relieve repression of the *P15INK4B* gene. Investigating the dynamic DNA methylation of the *P15INK4B* gene revealed that transcriptional silencing of the *P15INK4B* gene by the ZNF217/CoREST complex involves promoter hypermethylation, which is mediated, at least in part, by the recruitment of the DNA methyltransferase 3 alpha (DNMT3A) [28]. In contrast, active demethylation of the *P15INK4B* gene was induced by TGF-β/SMAD pathway and involved the loss of ZNF217/CoREST/DNMT3A in *P15INK4B* promoter [28]. ZNF217 overexpression was sufficient to prevent the TGF-β-dependent program by impairing the recruitment of factors allowing active demethylation at *P15INK4B* gene. Altogether, these studies highlighted the key role of ZNF217 in the maintenance of dynamic DNA methylation/demethylation of the *P15INK4B* gene. By refining the DNA methylation status at *P15INK4B* gene; ZNF217, thus, impairs proliferative control in cancer cells. Given the importance of DNA methylation in gene expression regulation, further studies would be required to elucidate whether the co-operation between ZNF217 and CoREST/DNMT3A is also involved in the epigenetic regulation of key genes implicated in ZNF217-dependent programs. Finally, as ZNF217-ΔE4 isoform was associated with deregulated DNA methylation of the *ZNF217* gene at key CpG sites [38], it would be important to decipher whether the ZNF217-ΔE4 isoform may co-operate with the DNA methylation/demethylation machinery.

## 4. ZNF217/Zfp217 Regulates the Epitranscriptome to Promote Cell Reprograming and Pluripotency

Methylated adenine residues are present in a large subset of RNAs, such as messenger RNAs (mRNAs) and long noncoding RNAs (lncRNAs), referred to as the epitranscriptome. M^6^A modification is dynamic and reversible, as with other epigenetic modifications. For mRNAs, N6-methyladenosine (m^6^A) is the most abundant biochemical modification observed and is involved in RNA metabolism, including mRNA instability, as well as translation, splicing, export and folding. M^6^A is accomplished by “writer” proteins belonging to a methyltransferase complex involving the methyltransferase-like 3 catalytic subunit (METTL3) and other accessory subunits, such as METTL14. Conversely, m^6^A “erasers” remove m^6^A modifications through the action of a family of demethylases, such as the alpha-ketoglutarate-dependent dioxygenase AlkB Homolog 5 (ALKBH5) or the FTO. Finally, “m^6^A readers”, such as members of the YTH domain-containing protein family (YTHDF), recognize and bind m^6^A to implement the corresponding function. For example, YTHDF2 is involved in the regulation of m^6^A-associated mRNA stability, and knockdown of YTHDF2 stabilizes mRNA possessing YTHDF2 binding sites [47]. Altogether, writers, erasers and readers co-ordinately shape the cellular epitranscriptome (for review [48,49]). Emerging evidence indicates that aberrant m^6^A modification is closely associated with cancer and involved in cancer progression (for review [48,49]). Furthermore, downregulation of METTL3, upregulation of ALKBH5 or YTHDF2 dysregulation are associated with poor prognosis in several types of cancers [48,50].

Recent studies evidenced that ZNF217/Zfp217 co-ordinates epigenetic and epitranscriptomic regulation through several crucial regulatory functions in m^6^A deposition. The pioneer and elegant study from Aguilo and collaborators demonstrated that Zfp217 binds to the m^6^A catalytic methylase METTL3 (but not to METTL14), sequesters METTL3 and, thus, restricts the METTL3-driven m^6^A deposition at key mRNAs in embryonic stem cells (ESCs), leading to increased mRNA stability and expression levels (Figure 2) [12]. Two further studies observed that silencing or overexpressing *Zfp217* inversely correlates with *METTL3* expression levels [13,51], while not observed in the original work [12]. In rat models, mRNA levels of the m^6^A demethylase *FTO* were co-ordinately regulated by *Zfp217* overexpression or silencing [13], and positive correlation between *Zfp217* and *FTO* expression levels existed [52]. Mechanistically, the *FTO* gene is a direct target for the Zfp217 TF, which positively regulates the transcription of *FTO* after binding to its promoter (Figure 2) [13]. The same study, conducted on mouse fibroblasts also highlighted that Zfp217 physically binds to and sequesters the YTHDF2 reader to keep the accessibility of FTO to m^6^A sites (Figure 2) [13]. Altogether, those findings support that, besides its TF function, Zfp217 also controls gene expression by shaping the epitranscriptome on target mRNAs, through the regulation of m^6^A writers, erasers or readers. Accordingly, a recent study investigating the expression and prognostic value of m^6^A regulators expression levels in human hematologic malignancies identified *METTL3*, *ZNF217* and *ALKBH5* as candidate oncogenic genes for hematologic system tumors [53].

Pluripotency is the first molecular program reported to be regulated through Zfp217-driven control of m^6^A deposition at key mRNAs [12,23,25,54]. Indeed, Zfp217 controls the epitranscriptome of key pluripotency genes, modulating murine ESCs reprograming [12]. By sequestering METTL3, Zfp217 negatively affects m^6^A methylation of the core pluripotency factors *NANOG*, *SOX2*, *KLF4* and *MYC* mRNAs, thus protecting those transcripts from degradation [12]. Meantime, Zfp217 binds to the promoter and enhancer regions of *NANOG, SOX2*, *KLF4* and *MYC* genes, leading to direct and positive transcription of those core pluripotency factors [12]. Altogether, Zfp217 tightly pairs direct gene transcription and m^6^A mRNA methylation restriction of pluripotency genes to maintain ESC renewal and somatic cell reprograming (Figure 2).

BCa stem cells (BCaSCs) are a subpopulation of cells within the tumor with dual properties of self-renewal and differentiation. Cancer stem cells are key actors of tumor initiation and progression [55]. BCaSCs are resistant to chemotherapy, thus representing candidate residual cells at the origin of recurrent metastatic tumors [56]. The BCaSCs phenotype is maintained by the expression of the core pluripotency factors OCT4, KLF4, SOX2 and NANOG [57]. Supportive to Aguilo et al.’s work, a further study highlighted that hypoxia induces BCaSC specification through HIF-dependent overexpression of ZNF217, ALKBH5, pluripotency factors and through the negative regulation of m^6^A RNA methylation [25]. ZNF217, by interacting with METLL3, prevents METLL3-catalyzed m^6^A deposition at *KLF4* and *NANOG* mRNAs, while ALKBH5 demethylates m^6^A *KLF4* and *NANOG* mRNA. *NANOG* and *KLF4* mRNAs are, thus, prevented from degradation, leading to increased expression of KLF4 and NANOG proteins, which specify the BCaSC phenotype.

Melatonin has been shown to emphasize the pluripotency of ESCs *in vitro* and to recover the *in vivo* differentiation potency of long-term cultured ESCs [23]. Following melatonin/melatonin receptor 1 (MT1) interaction, low m^6^A deposition is observed, in particular to *NANOG*, *SOX2*, *KLF4* and *MYC* mRNA, leading to increased mRNA stability [23]. Mechanistically, melatonin exerts its impact on m^6^A methylation through Zfp217, after upregulating *Zfp217* expression levels in an MT1-dependent mechanism, leading to the JAK2/STAT3 pathway activation (Figure 2) [23]. Altogether, these studies converged to the statement that ZNF217/Zfp217 prevents m^6^A deposition to enhance pluripotency reprograming [12,23,25,54].

Conversely, a previous study reported that the Zfp217 TF, in co-operation with chromatin remodeling proteins, orchestrates the variability for ESC lineage commitment, resulting in escape from pluripotency [58]. The C-terminal binding protein 2 (CtBP2) preoccupies regions in active ESC genes and balances H3K27 acetylation and appropriate H3K27me3 levels at these loci during exit from pluripotency [45]. By regulating the epigenetic states of H3K27, Zfp217 has been demonstrated to facilitate CtBP2-mediated repression on active ESC genes (*NANOG* and *OCT4*) during differentiation to ultimately lead to the proper exit from pluripotency [58].

In conclusion, ZNF217/Zfp217 plays apparent opposite roles in regulating pluripotency, which may result from the fine-tuning of the expression of pluripotency factors at the epitranscriptomic level, at the direct transcriptional level or at the chromatin remodeling level.

## 5. Noncoding RNA Networks Target *ZNF217* mRNA, Control *ZNF217* Expression Levels and Govern ZNF217-Driven Functions

The tremendous progress of whole genome sequencing methods and transcriptome analysis has changed our perception of the untranslated transcripts named ncRNA and led scientists to investigate their functional regulatory activities in various cellular processes [59,60]. MicroRNAs (miRNAs) have been the most extensively studied in cancer and their identification, biogenesis and activity are, by far, the best known among ncRNAs [61]. Currently, the miRBase database (https://www.mirbase.org/index.shtml (accessed on 27 April 2022)) predicts ~2883 human miRNAs, and miRNA editing events are evolutionarily conserved in a range of mammalian and nonmammalian species [62]. The mature miRNA interferes with target expression by binding to complementary sequences in the mRNA 3’UTR or coding regions, inducing mRNA degradation or translational repression. Given the wide quantity and diversity of targets for each miRNA and despite the computational *in silico* prediction of these targets, it is really challenging to predict the effects of loss and/or gain of function for a given miRNA in cancer. Indeed, the miRNA complex interaction with mRNAs and other ncRNAs is context- and cell-type-dependent, allowing miRNAs to target oncogene-promoting and tumor-suppressor transcripts [63,64].

LncRNAs represent the largest class of noncoding transcripts (~55,000 human genes) and participate in gene expression regulatory networks through several functions [65,66]. LncRNAs are known (i) to remove TFs and regulatory proteins from chromatin, (ii) to regulate chromatin organization and DNA methylation in both cis and trans genes and stimulate the formation of ribonucleoprotein complexes to induce histone modification, and (iii) to sequester or “sponge” other RNA molecules, such as miRNAs. In this case, lncRNAs function as competing endogenous RNAs (ceRNAs) that absorb miRNAs, resulting in the upregulation of those miRNA-direct mRNA targets. LncRNAs are usually weakly expressed and poorly conserved among species when compared to miRNAs [66]. However, this weakness in phylogenic conservation would contribute to the identification and validation of the lncRNAs–oncogenic network by facilitating the functional association of the specific deregulated molecular species with target genes, making lncRNAs attractive molecules to decipher certain molecular mechanisms in cancer. Indeed, whole transcriptome sequencing of the Cancer Genome Atlas identified lncRNAs that were downregulated in BCa tissues, correlated with favorable prognosis and that functioned as tumor suppressors [67].

Circular RNAs (circRNAs) have emerged as a new type of molecule with intriguing molecular functions and there exists at least some relation between specific circRNAs and cancer features [68]. CircRNAs form a covalently closed continuous loop, are conserved across species, and thousands of circRNAs have been identified in Eukaryotes with tissue-specific, cell-specific and developmental-stage-specific expression patterns [68]. An increasing body of research suggests that circRNAs might be important modulators of tumor progression through the regulation of cell growth, migration, apoptosis and cell cycle [68]. CircRNAs possess various modes of action in regulating cellular functions, which include behaving as a ceRNA that can sponge miRNA [69].

Cumulating recent observations indicated that the expression and the activity of the oncogenic TF ZN217 are modulated by complex ncRNA networks. By questioning putative miRNA-*ZNF217* interaction using MirWalk2 tool [70], Xiang and colleagues identified 172 candidate miRNAs targeting *ZNF217* mRNA, of which 42 were conserved between human and mice [71]. A recent study [72] aiming at exploring the upstream regulatory miRNAs for *ZNF217* in BCa questioned several databases, which revealed six common candidate miRNAs also present in the 42 miRNAs list from Xiang and colleagues [71]. Here, we are only reviewing the literature data reporting miRNA formally validated to bind *ZNF217/Zfp217* mRNA and to affect ZNF217/Zfp217-governed molecular events. Several of those validated miRNAs were themselves sponged by specific lncRNAs or circRNAs, thus unraveling intricate ceRNA/miRNA/ZNF217 axes controlling ZNF217 expression levels and downstream ZNF217-dependent functions.

### 5.1. ZNF217-Driven EMT Could Be Epigenetically Regulated by Specific miRNAs and Intricate lncRNA Networks

ZNF217 governs complex intracellular tumorigenic networks, which are pro-metastatic circuits [1]. ZNF217 is part of the TFs known as major promoters of the EMT in BCa progression [9,10]. ZNF217-driven EMT incorporates several leverages (Figure 3). First, as previously mentioned, the *CDH1/E-cadherin* gene is a ZNF217-direct target, and *CDH1/E-cadherin* transcription is negatively regulated by a repressive transcriptional complex orchestrated by ZNF217 and involving LSD1 [4,29]. Second, the TGF-β pathway is a major driver of the ZNF217-induced features of EMT [9]. ZNF217 drives the activation of a TGF-β autocrine loop by directly promoting the transcription of *TGFB2* or *TGFB3* genes. Finally, ZNF217-induced EMT is paired with increased mRNA levels of TFs known as major inducers of EMT, such as *ZEB1* [9].

Accumulating data evidenced that deregulated ncRNA networks involving miR-200 family and lncRNA-ATB (lncRNA activated by TGF-β) are important regulators of ZNF217-driven EMT and TGF-β signaling in cancer. The miR-200 family represses EMT and tumor invasion by targeting the 3′UTRs of major drivers of EMT, including *TGFB* [73], *ZEB1* [73,74,75] and *ZNF217 (*Figure 3*)* [74,76]. The direct miR-200c/*ZNF217* and miR-200c/*ZEB1* interactions led to decreased expression levels of *ZNF217* and *ZEB1* and to the suppression of malignant phenotypes [74,76]. Additional complexity is brought by a feedback loop whereby the ZEB1 TF binds to the promoter of the *miR-200c* gene and transcriptionally represses miR-200c expression, thus leading to an inhibitory loop stabilizing EMT and promoting invasion of epithelial-derived tumors (Figure 3) [73].

The lncRNA-ATB was first identified in hepatocellular carcinoma (HCC), where it is upregulated and activated by TGF-β [77]. In HCC, lncRNA-ATB binds and sponges the miR-200 family, and promotes EMT, cell invasion and metastatic organ colonization [77]. In keloid fibroblast progression, the oncogenic lncRNA-ATB is activated and sponges miR-200C, allowing ZNF217-activated autocrine TGFB2 secretion (Figure 3) [76]. In PCa, lncRNA-ATB promotes EMT paired with increased mRNA and protein levels of both ZNF217 and ZEB1 [78]. Finally, high lncRNA-ATB expression levels were associated with poor prognosis both in HCC and PCa [77,78].

Altogether, a signaling axis consisting of lncRNA-ATB/miR-200c/ZNF217/TGF-β2/ZEB1 participates in EMT and tumor progression. Briefly, the miR-200 family targets *ZNF217*, *ZEB1* and *TGFB2* mRNAs, and the ZNF217 protein upregulates autocrine TGF-β signaling to transcriptionally activate *ZEB1*, which itself could then exert a feedback inhibition (at least on miR-200c). TGF-β signaling activates lncRNA-ATB, which acts as a ceRNA to sponge miR-200c, resulting in the upregulation of *ZEB1* and *ZNF217* expression levels and induction of EMT.

MiR-141 belongs to the miR-200 family and downregulates EMT and malignant phenotypes, particularly by directly binding to the 3′UTRs of *TGFB2* mRNA (Figure 3) [73]. Conversely, the EMT driver ZEB1 directly binds to the promoter of the *miR-141* gene, thus negatively regulating the transcription of miR-141 (Figure 3) [73]. A recent study highlighted that miR-141-3p suppressed the activation of the TGF-β/Smad2 signaling pathway by targeting the 3′-UTR of *ZNF217*, whereas the lncRNA MALAT1 sponges miR-141-3p, thereby promoting the activation of the ZNF217/TGF-β2/SMAD2 axis (Figure 3) [79].

Apart from the miR-200 family, Xu and colleagues [72] recently discovered that miR-135 exerts inhibitory impact on EMT initiation, cell migration and cell invasion in BCa cells by directly silencing *ZNF217*. Accordingly, in BCa tumors with lymph node metastasis, an inverse correlation between *ZNF217* mRNA and miR-135 expression levels exists, showing low miR-135 expression levels associated with high *ZNF217* expression levels [72]. The *NANOG* gene encodes for a stemness protein able to promote EMT [80]. As described above, Aguilo and others discovered that ZNF217/Zfp217 drives *NANOG* mRNA stabilization/upregulation by negatively regulating its m^6^A methylation status (Figure 2 and Figure 3) [12,25,54]. Furthermore, *in vitro* and *in vivo* experiments highlighted that miR-135 impedes EMT in BCa cells and suppresses *in vivo* tumor growth and metastasis *via* disruption of the ZNF217/NANOG axis [72]. Mechanistically, miR-135 indirectly enhanced m^6^A modification at *NANOG* mRNA by negatively targeting *ZNF217* mRNA.

MiR-503, known to act as a tumor suppressor in several human cancers, inhibits PCa growth *in vitro* and *in vivo* [81]. Previous studies highlighted that the upregulation of ZNF217 and decreased expression levels of specific miRNAs targeting *ZNF217* (such as miR-24) promoted the growth of PCa cell lines [82]. MiR-503 directly targeted *ZNF217* in PCa cells, and *ZNF217* expression levels negatively correlated with miR-503 expression levels in advanced PCa tissues [81]. Patients with lower levels of miR-503 and higher *ZNF217* expression in tumors had significantly shorter survival time than those who had higher miR-503 and lower *ZNF217* expression levels. ZNF217 overexpression could overcome the repressed cell proliferation, migration and invasion induced by miR-503 in PCa cell lines, whereas knockdown of *ZNF217* mimicked the tumor-suppressive effects induced by miR-503 overexpression. Finally, the study demonstrated that the GATA3 TF binds to *miR-503* promoter and positively regulates its transcription, leading to the negative regulation of ZNF217 downstream EMT genes [81]. Altogether, a GATA3/miR-503/ZNF217 axis participates in controlling PCa progression (Figure 3).

In HCC, silencing ZNF217 inhibited cell proliferation *in vitro* and tumor growth in mice xenograft, whereas its enforced expression promoted EMT and HCC cell invasion [83]. HCC progression was driven by ZNF217 after mobilizing LSD1 at the *CDH1/E-cadherin* promoter, leading to *in situ* depletion of H3K4me2 and decreased *CDH1/E-cadherin* transcription. The same study showed that *ZNF217* mRNA is a direct target for miR-101 (Figure 3) [83], a well-reported tumor suppressor in HCC [84]. ZNF217 is a functional mediator of the inhibitory effects of miR-101 on the HCC progression, highlighting a miR-101/ZNF217/CDH1 axis contributing to HCC aggressiveness [83]. High mRNA levels of *ZNF217* or *LSD1* were associated with bad prognosis in HCC tumor patients, while high miR-101 levels were associated with better overall survival [83], supporting the clinical relevance of the findings.

The lncRNA prostate-associated transcript 1 (PCAT1) is a well-known oncogene that was found to be upregulated in colorectal cancer tissues and exosomes derived from colorectal patients [85]. The same study identified that *ZNF217* expression levels were negatively regulated by lncRNA PCAT1 silencing. The study suggested that lncRNA PCAT1 might co-ordinate ZNF217 to enhance colorectal cancer progression through the regulation of an EMT axis involving two metastasis-associated 1 family members (MTA2/MTA3) and Snai1/E-cadherin (Figure 3) [85]. While the authors argue a putative physical interaction between lncRNA PCAT1 and *ZNF217*, future work is further needed to confirm and to unravel the impact of any possible physical interaction between *ZNF217* and any specific lncRNA.

### 5.2. Trastuzumab Resistance Is Mediated by a lncRNA-ATB/miR-200c/ZNF217/TGF-β Axis

Trastuzumab (Herceptin™) is a human monoclonal antibody directed against the epidermal growth factor 2 (HER2) and is successfully used for therapy of early-stage and metastatic HER2-positive (HER2+) BCa. Meanwhile, the development of trastuzumab resistance (TR) and distal metastasis, mostly within a year, are the leading causes of mortality in HER2+ BCa. Accumulating data give evidence of a strong link between EMT and drug resistance [86]. Additionally, the EMT regulator TGF-β signaling is sufficient to promote multiple-drug resistance in various malignancies [87]. Preclinical BCa models for acquired TR displayed decreased expression levels of miR-200c, associated with higher mRNA levels of two of its direct targets, *ZNF217* and *ZEB1* [74,88]. In TR BCa cell lines and in tissues of 50 TR BCa patients, high expression levels of lncRNA-ATB, which sponges miR-200c, were detected and were inversely correlated with miR-200c expression levels but positively correlated with *ZNF217* and *ZEB1* mRNA levels [88]. Silencing miR-200c or upregulating lncRNA-ATB accounted for high invasiveness and TR in BCa cells [74,88]. Conversely, silencing *ZNF217* or *ZEB1* in TR BCa cells repressed cell invasion and restored, at least partially, trastuzumab sensitivity [74]. Altogether, in TR BCa cells, lncRNA-ATB promotes TR by competitively binding to miR-200c, leading to the upregulation of *ZEB1* and *ZNF217*, and then inducing EMT [88], while miR-200c inhibition is reinforced by nested feedback circuits governed by miR-200c/ZEB1 and miR-200c/ZNF217/TGF-β/ZEB1 [74]. Re-expression of miR-200c or silencing *ZNF217* and *ZEB1* could, thus, shift and rebalance regulatory circuits associated with the reversion of TR and metastasis in HER2+ BCa cells [88]. Supporting *in vivo* data indicated that re-expressing miR-200c is effective in restoring trastuzumab sensitivity in a xenograft BCa model and in inhibiting the *in vivo* development of metastasis of TR BCa cells [74]. Deciphering those complex regulatory networks could, therefore, delineate future candidate therapeutic strategies to prevent or bypass TR resistance in HER2+ BCa.

### 5.3. Tumor Progression Is Accelerated by Oncogenic LncRNAs or Oncogenic circRNAs Sponging miRNA and Upregulating ZNF217 Expression

In addition to the two aforementioned axes (lncRNA-ATB/miR-200c/ZNF217/TGF-β2 and MALAT1/miR-141-3p/ZNF217/TGF-β2), both involved in the activation of the TGF-β signaling pathway and EMT, ZNF217 is part of additional intricate ceRNA-driven regulatory circuits involved in tumor progression (Figure 4A). The small nucleolar RNA host gene (SNHG) family is a group of transcripts that can be processed into small nucleolar RNAs and may exert oncogenic functions. In non-small-cell lung cancer (NSCLC), lncRNA SNHG15 expression levels were significantly higher in tumor tissues compared to adjacent tissues, higher in patients with stage III–IX compared to those with stage I–II, and associated with lower five-year survival rates [89]. The same study highlighted that lncRNA SNHG15 absorbed miR-211-3p and an inverse pattern of expression of those two protagonists exists in NSCLC. Overexpression of miR-211-3p could rescue lung cancer cells from lncRNA SNHG15 activation of cell proliferation and migration. *ZNF217* mRNA was further demonstrated to be a direct target for miR-211-3p, and ZNF217 overexpression was sufficient to reverse the inhibitory impact of miR-211-3p on the proliferation or the migration of lung cancer cells [89]. Altogether, lncRNA SNHG15 promoted NSCLC progression by sponging miR-211-3p and through the lncRNA SNHG15/miR-211-3p/ZNF217 axis. Another member of the SNHG family, lncRNA SNHG1, has recently been shown to contribute to the progression of breast or lung cancers [90,91]. Interestingly, a lncRNA SNHG1/miR-361-3p/ZNF217 axis has been identified in a model of Alzheimer disease, where lncRNA SNHG1 absorbs miR-361-3p and where *ZNF217* mRNA is a direct target for miR-361-3p [92]. Future work is, thus, needed to decipher whether lncRNA SNHG1/ZNF217 regulatory circuits also participate in tumor progression.

In cervical cancer, Yang and colleagues discovered that *ZNF217* is targeted by miR-3163 and that the lncRNA CTBP1 antisense RNA 2 (CTBP1-AS2) regulates cancer progression after miR-3163 sponging to upregulate *ZNF217* [93]. *In vitro* rescue assays confirmed that ZNF217 overexpression reverses the antitumor impact of CTBP1-AS2 silencing (reversion of G0/G1 cell cycle arrest, of the proapoptotic function and of the inhibition of cell migration and cell invasion). ZNF217 overexpression, thus, recapitulated the oncogenic function of lncRNA CTBP1-AS2. Supportive *in vivo* experiments validated the oncogenic role of the CTBP1-AS2/miR-3163/ZNF217 axis in accelerating tumor growth. In cervical tumor samples, high CTBP1-AS2 expression levels, low miR-3163 expression levels or high *ZNF217* mRNA levels were associated with unfavorable prognosis. Altogether, this study was pioneering in demonstrating the oncogenic role of CTBP1-AS2 in cancer progression and in discovering a new CTBP1-AS2/miR-3163/ZNF217 regulatory network. Since this study, recent works conducted in other cancers validated the oncogenic function of lncRNA CTBP1-AS2 [94,95,96]. Importantly, one study discovered that lncRNA CTBP1-AS2 was able to modulate the TGF-β/SMAD2/3 pathway [94]. Altogether, it is, thus, tempting to speculate that ZNF217 contributes to the CTBP1-AS2-mediated TGF-β pathway activation through a CTBP1-AS2/ZNF217 axis.

Another work conducted on epithelial ovarian cancer (EOC) identified a novel lncRNA OIP5 antisense RNA 1 (OIP5-AS1)/miR-137/ZNF217 regulatory circuit that accelerates ovarian cancer progression [97]. LncRNA OIP5-AS1 absorbed miR-137, and *ZNF217* mRNA was directly targeted by miR-137. The lncRNA OIP5-AS1 exerted pro-tumorigenic function in EOC cells *in vitro* and tumor growth *in vivo*. In EOC cells, ZNF217 overexpression was able to counteract the suppressive impact of silencing OIP5-AS1 on cell proliferation, migration, invasion and EMT. Altogether, ZNF217 rescued the oncogenic function of the lncRNA OIP5-AS1, and OIP5-AS1 aggravated EOC tumor progression by sponging miR-137 and upregulating ZNF217. Supporting data in BCa reported that high lncRNA OIP5-AS1 expression levels [98], similarly to increased ZNF217 expression levels [9,37], are associated with poor prognosis.

Keloids are considered to be benign skin tumors, with uncontrolled fibroblast proliferation and low remission rate [99]. By positively regulating *ZNF217* expression after decoying miR-182-5p, the lncRNA HOXA11-AS regulates the miR-182-5p/ZNF217 axis to induce the proliferation and migration of keloid fibroblasts *in vitro*, while promoting keloid formation and growth in mouse [100].

CircRNAs are emerging ncRNAs and much effort is needed to unravel the potential importance of circRNAs in cancer. In gastric cancer (GCa), a new network has been discovered where circRNA casein kinase 1 gamma 1 (circCSNK1G1) modulated the miR-758/ZNF217 axis to promote GCa progression [101]. High levels of circCSNK1G1 were present in GCa tissues compared to normal tissues, and high expression levels of circCSNK1G1 in GCa tumor samples were significantly associated with shorter survival. In mice experiments, silencing circCSNK1G1 restrained tumor growth. The study further discovered that circCSNK1G1 sponges miR-758, miR-758 directly binds to the 3′-UTR of *ZNF217* mRNA, and ZNF217 is a downstream effector of the circCSNK1G1/miR-758 axis in regulating GCa progression. A study recently highlighted that another circRNA, named hsa-circ-00690094, also increases *ZNF217* expression levels through the sponging of miR-758-3p [102] and, thus, functions as an oncogene promoting BCa progression. In addition to circCSNK1G1 and hsa-circ-0069094 circuits, ZNF217 seems to be part of additional axes involving circRNAs, supported by the discovery of the new circLPAR1/miR-212-3p/ZNF217 regulatory circuit in neuronal cells [103]. As recent studies indicated that deregulated expression levels of circLPAR1 are associated with tumorigenesis and poor prognosis [104,105], future work is needed to decipher any oncogenic function of the circLPAR1/miR-212-3p/ZNF217 axis.

All the above-mentioned studies emphasized that ZNF217 is a key downstream effector of several oncogenic lncRNA (lncRNA-ATB, MALAT1, SNHG15, CTBP1-AS2 and OIP5-AS1) or oncogenic circRNA (circCSNK1G1) in promoting tumor progression (Figure 4A). On the other hand, ZNF217 has been also reported to negatively regulate the expression of tumor-suppressive lncRNAs (Figure 4B). Pang and colleagues identified EPB41L4A-AS2 as a new tumor-suppressive lncRNA in BCa that triggered reduced BCa cell proliferation and invasion and increased apoptosis [67]. ZNF217 negatively regulates the expression of EPB41L4A-AS2 at the transcriptional level, most probably through the enhanced recruitment of EZH2 and increased H3K27me3 enrichment at EPB41L4A-AS2 locus [67]. A recent study identified a new tumor-suppressive lncRNA, namely LINC00111, whose transcription is negatively regulated by the TXB2 transcriptional complex [106]. In BCa cells, ZNF217 was identified to be a novel interactor of TBX2 that binds to 30% of TXB2-bound promoters, including that of the lncRNA LINC00111.

In conclusion, a complex and close interplay exists between ZNF217 and specific ncRNA regulatory axes where ZNF217 can act either as a downstream effector of oncogenic ncRNAs (lncRNA/circRNA) or as an upstream negative transcriptional regulator of tumor-suppressive lncRNAs.

## 6. ZNF217-Associated Epigenetic Events Interfere with BCa Risk Factors

### 6.1. Estrogen Exposure

Several risk factors conductive to the development of BCa are known (for review [107]). One of the most important are hormonal factors, mainly related to the time of exposure to estrogens, and procreative factors, including the number of children born, the age of birth of the first child, or breastfeeding. At the molecular level, estrogen-signaling pathways are the efficient targets for endocrine therapies because of their mitogenic role in BCa cells. The most predominant subtypes of BCa are the ER+ subtypes, representing 70–80% of diagnosed BCa. The canonical genomic activity of ERα involves activation upon estrogen binding, leading to nuclear translocation and direct or indirect transcriptional control of estrogen-related genes, thus mediating the mitogenic effects of estrogens. ERα is important in BCa, both as a good prognostic biomarker and as a therapeutic target for endocrine therapy. Our group previously discovered that ZNF217 is a transcriptional regulator that binds to ERα to amplify the estrogen response in BCa [37]. High *ZNF217* expression levels allowed the stratification of luminal-A BCa, were associated with poorer prognosis and were predictive of earlier relapse under endocrine therapy [37,108]. ZNF217 is upregulated in ER+ BCa compared to ER− BCa [37], while, paradoxically, estrogen exposure of the ER+ BCa cell line MCF-7 led to decreased *ZNF217* mRNA and ZNF217 protein levels [109].

In response to estrogens, ERα stimulates a transcriptional program involving both coding and noncoding RNAs [110]. Master regulators of estrogen response have been identified in estrogen-stimulated MCF-7 by integrative analyses of mRNA and miRNA profiles [111]. Experimental evidence led to the conclusion that the estrogen-induced miR-503 exerts tumor-suppressive properties in BCa cells, at least in part, by direct targeting of the 3′-UTR of the *ZNF217* mRNA [111]. In BCa tumor samples, high expression levels of *ZNF217* are associated with shorter relapse-free survival [9,37,108], while high miR-503 levels are associated with improved survival [111]. Interestingly, the same study found a decrease in both *GATA3* and miR-503 expression levels upon estrogen stimulation [111]. As previously mentioned, GATA3 positively regulates the transcription of miR-503, and a GATA3/miR-503/ZNF217 axis exists in cancer cells [81]. Altogether, those findings suggested an intricate estrogen-driven GATA3/miR-503/ZNF217 network to avoid ZNF217-induced attenuation of the estrogen response and to limit further stimulation of the estrogen response signaling.

### 6.2. Obesity and Adipogenesis

As a major global public health problem, obesity is a complex disease that is recognized as part of BCa risk factors. In obesity, excess body fat is largely stored through adipocyte hypertrophy and hyperplasia (cell number increase) mechanisms. The hyperplasia in obesity is the result of preadipocyte differentiation into mature adipocytes, namely, adipogenesis. Previous *in silico* analysis of transcriptomic data allowed building of a Bayesian network in the vicinity of the *ZNF217* oncogene, revealing a strong interference with genes involved in lipid metabolism [112]. Demonstration of Zfp217’s physiological roles in adipogenesis was further reported by Xiang and colleagues [71], and *in vivo* data highlighted that Zfp217 also functions as a regulator of systemic energy [113]. Zfp217 expression was positively correlated with adipocyte differentiation, consistently with adipogenic marker peroxisome proliferator-activated receptor gamma (PPARG) [71]. Zfp217 facilitated the adipogenesis process by co-operating with EZH2, which inhibits the transcription of Wnt signaling genes, such as *WNT6* and *WNT10B*, the latter being both negative regulators of adipogenesis [71]. Attempting to identify adipogenesis regulator miRNAs, an *in silico* analysis identified 42 common candidate miRNAs targeting the 3′-UTR of *ZNF217* and *Zfp217* mRNA, then refined the screening by gene ontology enrichment analysis. The study led to the identification and validation of three miRNAs (miR-503-5p, miR-135a-5p and miR-19a/b-3p) that were directly bound to *Zfp217* mRNA, downregulated *Zfp217* expression levels and functionally impaired adipocyte differentiation [71]. These findings indicated that Zfp217 possesses a role in adipogenesis by promoting differentiation of preadipocytes into mature adipocytes, and a specific set of adipogenic miRNAs governed this function.

Compiling observations showed that the versatile m^6^A modification affects cancer and obesity [114,115]. Converging data from two recent studies highlighted that Zfp217 regulates adipogenesis by coupling gene transcription to m^6^A mRNA modification (Figure 2) [13,51]. The m^6^A demethylase FTO is also known to possess an activity linked to adipogenesis. The Zfp217 binding to *FTO* promoter upregulates *FTO* expression, which decreases m^6^A modification and augments adipogenesis [13]. Furthermore, Zfp217 interacts with and sequesters the m^6^A “reader” YTHDF2, thereby facilitating the accessibility of FTO to m^6^A sites to facilitate adipogenic differentiation [13]. Another study reported that the depletion of Zfp217 in preadipocytes was paired with increased expression levels of the m^6^A methyltransferase METTL3, which upregulates the m^6^A levels of *CCND1/cyclin D1* mRNA [51]. The authors concluded that adipogenesis was, thus, regulated *via* a Zfp217 control of mitotic expansion in an METTL3-m^6^A-dependent manner [51]. Altogether, Zfp217 promotes adipogenic differentiation by orchestrating m^6^A epigenetic modification (by direct transcriptional regulation of m^6^A regulators or by direct binding to key m^6^A regulators), while *Zfp217* mRNA is a direct target for key miRNAs controlling adipogenesis.

### 6.3. Stroma Stiffness

High mammography density, which has been attributed to a higher epithelial density, associates with high life-time risk of malignancy [116]. Every 3–6% in mammography density is associated with a 10% higher risk of developing BCa [117]. A dense interstitial stroma is characterized by high levels of fibrillary and cross-linked collagen that increases the stiffness of the tissue stroma [118]. As stiffer breast tissue stroma increases epithelial density and reprograms the breast epithelium towards a pre-oncogenic state, it is necessary to unravel the molecular mechanisms sustaining breast tissue stiffness. An elegant study identified miR-203 and its direct target *ZNF217* as critical players in the extracellular matrix stiffness regulation in mammary tissue [119]. Stiffness and high collagen density repressed miR-203 expression that is found decreased in breast tissues with high mammography density. Mouse model experiments highlighted a causal interplay between stroma stiffness, decreased miR-203 expression levels, higher *Zfp217* expression levels, increased Akt activity and increased mammary epithelial proliferation. In breast tissue of women with high mammographic density, the epithelial proliferation and density were positively correlated with *ZNF217* expression levels and inversely with those of miR-203. Additionally, mir-203 targets roundabout guidance receptor 1 (*ROBO1*) mRNA, which codes for a well-known tumor suppressor, thus possessing opposite effects than those of the ZNF217 oncogene. Therefore, a subtle balance between *ROBO1* and *ZNF217* expression levels may govern the stroma stiffness and may orchestrate the fate of the mammary epithelium for malignant transformation. Supporting and preliminary observations also indicated that the *ZNF217/ROBO1* mRNA ratio was increased in 24% of the high-mammography-density human breast tissues [119]. Altogether, this study illustrated how stiff stroma may enhance BCa risk by activating ZNF217 in a miR-203-dependant mechanism.

## 7. NcRNAs Networks Targeting ZNF217 Are Also Involved in Programing Cancer Cells to Metastasize to Bone

Bone metastases (BM) represent a common complication of cancer, whose incidence reaches up to 65–90% in PCa and about 65–75% in BCa [120]. PCa-derived BM are prevalently osteoblastic, while BCa-derived BM exhibit a prevalent osteolytic pattern. A fine balance exists between osteogenesis and bone resorption, with clear dysregulation during the development of BM. Bone turnover is the result of converse activities of osteoblasts and osteoclasts and is regulated by several factors (cytokines, hormones, vitamins, etc.) [121]. Both cancer and bone cells secrete factors that interplay in a process named “vicious cycle” of BM. Cancers with BM are essentially incurable with the current anticancer therapies, but these therapies have demonstrated to improve outcome. The mechanisms driving the preferential spread of cancer cells to the bone remain to be deciphered, and only few biomarkers and mediators of the bone metastatic process in BCa cells have been identified so far [122]. In order to prevent bone metastases, there is an urgent need to identify oncogenic drivers and regulators of metastasis genes that predispose these subsets of tumor cells to disseminate in the bone marrow.

We previously discovered that ZNF217 expression was an early indicator of BM and that the ZNF217 oncogene belongs to the few demonstrated key drivers of BM in BCa [123,124]. Patients with high *ZNF217* mRNA expression levels in primary breast tumors had a higher risk of developing BM. ER+ breast tumors are known to preferentially spread to bone, with postmenopausal, hormono-sensitive (ER+/Luminal) BCa patients having a high risk of bone recurrence. The predictive value of *ZNF217* expression levels for BM development was the most powerful in ER+/Luminal BCa [123]. Ectopic ZNF217 overexpression conferred highly aggressive properties to human BCa cells *in vivo*, promoting early onset of osteolytic lesions in the skeleton of animals. Ectopic ZNF217 expression in BCa cells was associated with constitutive activation of the bone morphogenetic protein (BMP) pathway and increased expression of downstream effectors, such as *RUNX2* [123]. BCa cells overexpressing ZNF217 are committed in a close crosstalk with osteoblasts and osteoclasts: soluble factors released by differentiated osteoblasts stimulated chemotaxis of ZNF217-overexpressing BCa cells to the bone environment and ZNF217-overexpressing BCa cells secreted soluble factors that promote osteoclast differentiation, which was in accordance with the observed osteolytic lesions in animals.

Accumulating data evidenced that ncRNAs are functionally involved in the regulation of bone remodeling, of BM development, and provide new biomarkers of bone-homing malignancies [125,126,127,128]. Here, we are emphasizing that several miRNAs directly targeting *ZNF217* (miR-203, miR-135 and miR-141-3p) or lncRNA indirectly regulating *ZNF217* expression levels (MALAT1 and HOXA11-AS) are also involved in the development of BM program in PCa, BCa or NSCLC.

MiR-135 and miR-203 directly target and downregulate *ZNF217* expression levels [72,119]. Opposite to the ZNF217′s promotion of BM development, miR-135 and miR-203 possessed BM suppressive function. Ectopic expression of miR-203 in PCa cells prevented BM *in vivo* by inhibiting EMT, while miR-203 expression levels were downregulated in bone metastatic PCa [129,130]. In PCa cells, miR-203 directly targets *RUNX2* mRNA, a downstream effector of the BMP pathway and a promoter of BM progression [129]. *RUNX2* and *SMAD5*, the latter being also a member of the BMP pathway, are also directly bound by miR-135 [131]. *In vivo* experiments demonstrated that the ectopic expression of miR-203 and miR-135 in bone tropic BCa cells reduced the development of osteolytic bone metastases by targeting *RUNX2* [131]. Altogether, miR-203 and miR-135 are both onco-suppressors of BM progression and direct negative modulators of the BM driver ZNF217.

Low miR-141-3p expression levels have been involved in the progression and metastasis of several human cancer types. Huang and colleagues reported that miR-141-3p expression is downregulated in bone-metastatic PCa tissues compared with non-bone metastatic PCa tissues and is associated with serum prostate-specific antigen (PSA) levels, Gleason grade and BM status [132]. Overexpression of miR-141-3p suppressed the EMT, invasion and migration of PCa cells *in vitro*, and inhibited BM of PCa cells *in vivo* [132]. MiR-141-3p was also detected in exosomes produced by PCa cells and, after being taken up by recipient osteoblast precursors, demonstrated promotion of osteoblast differentiation, thereby facilitating sclerotic BM [133].

The lncRNA MALAT1 is significantly highly expressed in NSCLC tissues with BM and in NSCLC cell lines with high bone metastatic ability [134]. Increased expression of MALAT1 in PCa correlated with Gleason score, PSA, tumor stage and castration-resistant PCa [135]. The lncRNA HOXA11-AS regulates the osteotropism of PCa cells through specific downstream cytokine and integrin signal in osteoblastic cells [136]. Briefly, an HOXB13/HOXA11-AS axis regulates BM-related integrin and CCL2/CCR2 cytokine signaling in PCa cells, while paracrine action of exosomal HOXA11-AS secreted from PCa cells is able to modulate CCL2/CCR2 cytokine signaling in osteoblasts within the bone marrow milieu [136]. LncRNA MALAT1 and lncRNA HOXA11-AS promote the activation of the ZNF217-dependent functions after sponging miR-141-3p [79] and miR-182-5p [100], respectively. It is, thus, tempting to speculate that MALAT1/miR141-3p/ZNF217 or HOXA11-AS/miR182-5p/ZNF217 axes might exist in PCa tumors, contributing to the development of BM. Altogether, miR-205, miR-135, miR-141-3p, MALAT1 and HOXA11-AS may represent a new generation of predictive biomarkers and therapeutic targets in ZNF217-positive cancers prone to developing BM.

## 8. Future Directions

Recent technological advances offer a better understanding of the underlying epigenetic alterations during carcinogenesis and provide insight into the discovery of putative epigenetic biomarkers for detection, prognosis, risk assessment and disease monitoring. A remarkable difference between epigenetic changes and genetic alterations is that the former is reversible, making them attractive targets for therapeutic intervention. Unraveling epigenetic changes associated with cancer onset progression and metastasis development are, thus, also crucial to improve and delineate candidate strategies for cancer prevention and treatment.

Extensive data show that high *ZNF217* expression levels are associated with poor prognosis in several human cancers and predictive of response to chemotherapy or endocrine therapy (for reviews [1,2]). The present review provides an up-to-date report of the ncRNAs validated to modulate *ZNF217* expression levels and, consequently, ZNF217-driven functions. Interestingly, enrichment within extracellular vesicles, such as exosomes, and their subsequent delivery into recipient cells suggest possible roles for ncRNAs in modulating the tumor microenvironment to promote tumor growth and metastasis. NcRNAs could be released into body fluids after encapsulation in extracellular vesicles or in association with partners preventing their degradation [137], thus making them a new generation of exciting biomarkers for prognosis and therapeutic prediction in oncology. Strikingly, aberrant circulating expression of several ZNF217-modulating ncRNAs (lncRNA-ATB, SNHG15, OIP5-AS1, MALAT1, miR-200 family, miR-503, miR-24, miR-101, miR-361, miR-137, miR-212, miR-19a and miR-203) have been detected in body fluids and display candidate predictive or prognosis biomarker value [138,139,140,141,142,143,144,145,146,147,148,149,150,151]. Previous data also indicated that lack of methylation at the *ZNF217* locus in peripheral blood cells could predict BCa risk [39,40]. Altogether, besides assessing *ZNF217*/ZNF217 expression levels on tumor biopsies, assessing circulating lncRNAs/circRNAs/miRNAs targeting *ZNF217* or circulating DNA methylation status at *ZNF217* locus could represent a surrogate biomarker of poor prognosis or poor therapeutic response in cancer. It would also be important to consider the prognostic value of “miRNA-lncRNA/circRNA pairs” plasma levels or to combine DNA methylation and cell-free ncRNA biomarkers for improved monitoring of human cancers.

Accumulating studies have evidenced that epigenetic modifications contribute to the initiation and maintenance of EMT and their reversibility might explain the EMT plasticity and switch to mesenchymal–epithelial transition (MET) [152]. Multiple epigenetic alterations are well known to target *CDH1/E-cadherin* during EMT but, also, key EMT-inducing TFs (EMT-TF) [152,153]. Epigenetic regulation of EMT-TF could occur at several levels [152,153]: (i) the promoter of EMT-TF, such as *Twist*, is hypomethylated; (ii) EMT-TFs expression could be regulated by histone modification and chromatin remodeling; (iii) a multitude of miRNAs have been described to be involved in the epigenetic regulation of EMT-TF, such as *Snail*, *ZEB1*, *ZEB2* and *Twist1*; (iv) m^6^A deposition regulates EMT in cancer cells and translation of *Snail*. All those studies highlight the importance of fully understanding the wide diversity of epigenetic mechanisms regulating EMT-TF, such as ZNF217, with the ultimate aim to design therapeutic strategies counteracting tumor progression and metastasis development.

A tremendous quest in the development of so-called “epidrugs” (epigenetic drugs) recently emerged [27,154]. Although the use of epidrugs in cancer is still in its early stages, several anticancer therapeutic drugs targeting epigenetic modifiers, such as DNMT, EZH2 and HDAC, have been approved by the FDA and used in clinics; additional epigenetic drugs targeting chromatin modifiers are currently under clinical trial investigation (for review [27]). In parallel, using ncRNAs as therapeutic targets represents an alternative promising pharmacological strategy. Therapeutically targeting miRNAs in cancer aims at restoring their expression levels when downregulated or inhibiting their expression when upregulated. For this purpose, therapeutic approaches utilizing oligonucleotides are used [155], and some of them are already under Phase I/Phase II clinical trial investigation (e.g., NCT01829971, NCT02369198 and NCT03837457). Encouraging Phase I studies delivering a liposomal miRNA mimic in advanced solid tumors have emerged [156], but adverse effects accompanied these. Indeed, the main limitation of those therapeutic strategies resides in the improvement of a delivery system that would allow effective and safe transport of therapeutic nucleic acids into target cells after intravenous injection [155]. In comparison with miRNAs, there is much less experience with therapeutic targeting of lncRNAs, but current investigations are very similar to those used for miRNA therapeutics, since both are oligonucleotide-based strategies. Encouraging data showed that antisense oligonucleotides (ASOs) targeting specific lncRNAs have been proved to be efficient both *in vitro* and in mice experiments (for review [157]). An alternative strategy for lncRNA silencing describes the use of specific small molecules that interfere with the secondary and tertiary structures of specific lncRNAs [158]. An exciting recent study described the downregulation of the lncRNA HULC using a small molecule YK-4-279, leading to downregulation of the EMT driver *Twist1* expression levels after unleashing miR-186 and allowing its binding to *Twist1* [159]. Currently, the number of clinical trials with lncRNA-targeting strategies is increasing, and the FDA has approved a Phase I clinical trial in advanced metastatic cancer of Andes-1537, an ASO-targeting mitochondrial lncRNAs (NCT02508441). Finally, given the functional importance of m^6^A modification and regulators in human cancers, targeting m^6^A regulators is an attractive strategy for cancer therapy [160]. Pioneering recent studies highlighted that targeting dysregulated m^6^A regulators by small molecule inhibitors reveal exciting novel pharmacological strategies for anticancer therapies, alone or in combination with other therapeutics. As proof of concept, FTO inhibitors were demonstrated to possess anticancer properties both *in vitro* and in mice experiments, by inhibiting growth/survival and by sensitizing cancer cells to chemotherapeutic drugs (for review [160]).

## 9. Conclusions

The present review pinpoints that ZNF217 is part of complex ncRNAs circuits and a fine regulator of the epitranscriptome. Unraveling the close cross-talk between the ZNF217 oncogene and epigenetic mechanisms may lead to the identification of new prognostic biomarkers and the development of candidate anticancer therapies. While the oncogenic role of ZNF217 in several human cancers is well admitted, few ZNF217-targeting therapeutic options have emerged. Indeed, the complete ZNF217 3D structure is not available [6,161], limiting *in silico* drug design. To date, the Akt inhibitor triciribine is the only drug shown to counteract the ZNF217-driven deleterious effects *in vivo* and has been proposed as a clinical strategy to treat ZNF217-positive cancer patients [10,119,162,163]. Deciphering this close interplay between ZNF217 and epigenetic processes, thus, sets the stage for considering epidrugs strategies, alone or in combination with conventional treatments, and holds potent therapeutic potential for the treatment of ZNF217-positive cancers.

## Figures and Tables

**Figure 1 cancers-14-06043-f001:**
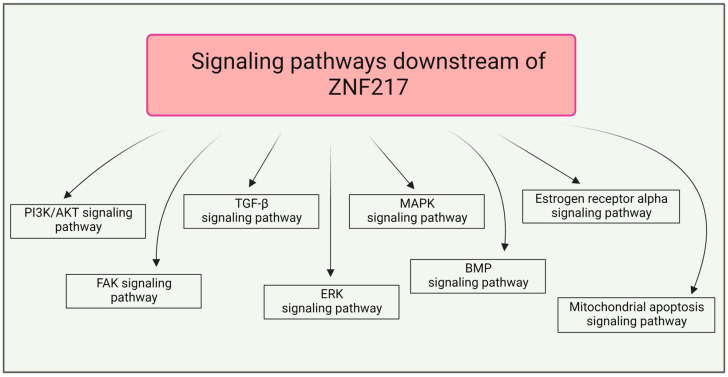
Signaling pathways regulated by ZNF217 and involved in ZNF217-driven functions in tumor progression. Created with Biorender.com.

**Figure 2 cancers-14-06043-f002:**
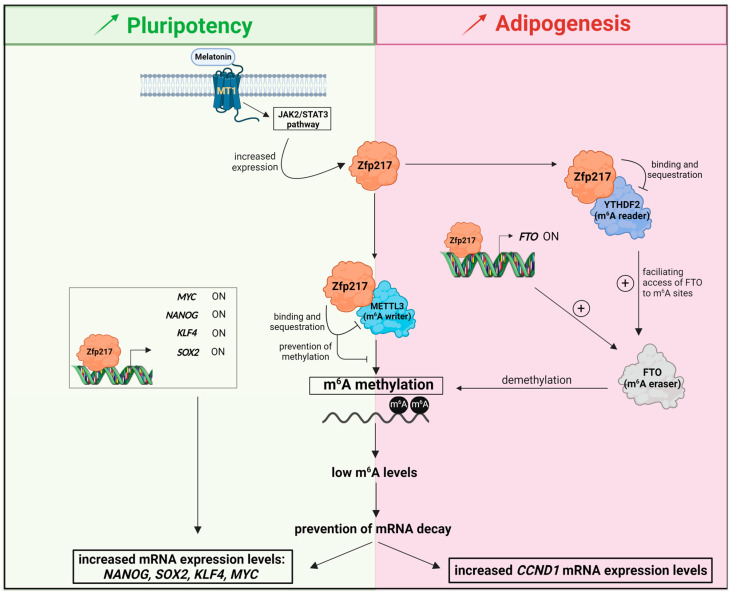
Zfp217 enhances pluripotency and adipogenesis by regulating the epitranscriptome. Created with Biorender.com.

**Figure 3 cancers-14-06043-f003:**
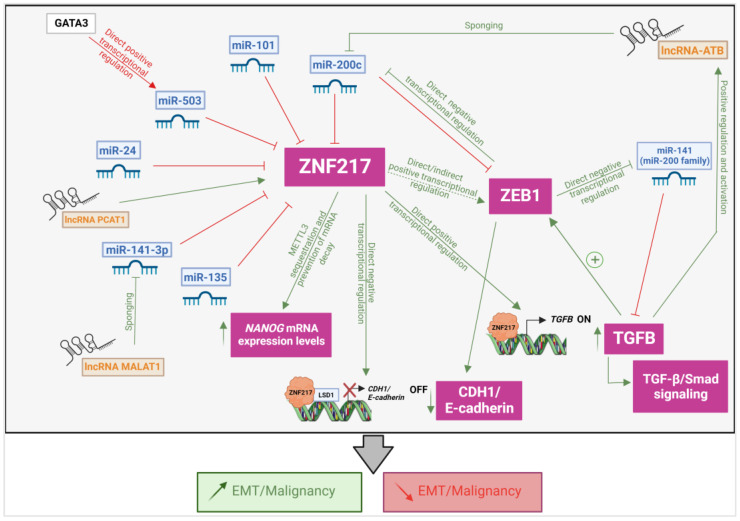
Intricate interactions between ZNF217 and ncRNAs regulate EMT and malignancy. Regulating events (miRNA binding to ZNF217 and other drivers of EMT, lncRNA sponging of miRNA, and transcriptional regulation) leading to activation of EMT and an increase in malignancy (arrows in red) or leading to inhibition of EMT and a decrease in malignancy (arrows in green). Arrows with bars refer to an inhibition. Created with Biorender.com.

**Figure 4 cancers-14-06043-f004:**
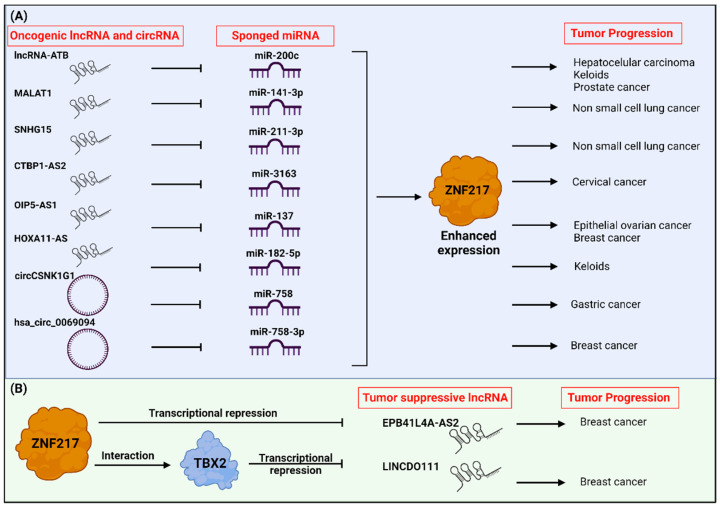
Interplay between ZNF217 and ncRNA promotes tumor progression. (**A**) lncRNA and circRNA sponge miRNA preventing their inhibition of ZNF217 and, thereby, leading to enhanced expression of ZNF217 that promotes tumor progression. (**B**) ZNF217 regulates the expression of tumor-suppressive lncRNAs through transcriptional repression. Created with Biorender.com.

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
