# Peer review of "The Intricate Interplay between the ZNF217 Oncogene and Epigenetic Processes Shapes Tumor Progression"

_cancers, 2022, doi:10.3390/cancers14246043_

Round 1

Reviewer 1 Report

This is an interesting review article related to the regulatory function of ZNF217 with a particular emphasis on its complex with ncRNA and its role as a fine regulator of epitranscriptome. The review represents a thorough account of the proposed mechanisms of action of Interplay between ZNF217 and ncRNA promotes tumor progression, in regulating EMT and malignancy. The authors have done well to collate the available information into a coherent and informative manuscript.  The length and level of detail is appropriate - it is neither too brief nor too overwhelming. The authors have cited the appropriate literature. The present review provides an up-to-date report of the ncRNA validated to modulate ZNF217 expression levels, and consequently, ZNF217 driven functions. Given the recent global focus on the regulation of the genome, chromatin remodeling and chromatin modification are very interesting and timely topics.  Therefore, this manuscript would be of interest to a wide readership, particularly, to those with an interest in chromatin biology and epigenetic mechanisms.
Overall, this manuscript is very well written, and the table and figures are excellent and highly informative.  In this reviewer's opinion this manuscript can be accepted for publication.

Author Response

Response : We thank the reviewer for his/her positive comment

Reviewer 2 Report

Fahme and Tien Le et.al., nicely summarized the diverse role of ZNF217 in tumor progression and described its molecular function in detail. The article is written well and contains enough (up-to-date) literature to back-up hypothesis and descriptions. Overall, this review summarized the accumulating evidence and conclusions of ZNF217, and I recommend its publication.

Minor concerns/suggestions

1.     What is known on posttranslational modifications of ZNF217?

2.     What distinguishes this paper from the recent review articles by Li et al., 2021? “ZNF217: the cerberus who fails to guard the gateway to lethal malignancy”. Please include one or two sentences about it.

3.     It would be great if the authors included a figure that summarized all of the interlink signaling pathways in cancer through ZNF217, regardless of cancer type.

Author Response

Response to Reviewer 2’s comments

Reviewer 2’s comments and suggestions for authors :

Comment 1 : Fahme and Tien Le et.al., nicely summarized the diverse role of ZNF217 in tumor progression and described its molecular function in detail. The article is written well and contains enough (up-to-date) literature to back-up hypothesis and descriptions. Overall, this review summarized the accumulating evidence and conclusions of ZNF217, and I recommend its publication.

Response to comment 1 :We thank the reviewer for his/her positive comment

Minor concerns/suggestions

Comment 2 :     What is known on posttranslational modifications of ZNF217?

Response to comment 2 : This is a very relevant and interesting question, as post-translational modifications might affect ZNF217/ZFP217 oncogenic function. To date, there is no study having ever investigated or validated the existence of any post-translational modification targeting the ZNF217/Zfp217 protein.  Aberrant expression levels thus represent the main mechanism associated with ZNF217 oncogenic function. As specified in our review (introduction section), little is known regarding TFs being formally demonstrated to be recruited at ZNF217/Zfp217 promoter and to positively regulate the transcription of this oncogene. Thus, elucidating the epigenetic mechanisms contributing to high ZNF217/Zfp217 expression levels and oncogenic functions is of utmost importance.

To fulfill the reviewer’s comment we have added a new sentence line 63-66 of the revised version of the manuscript.

Comment 3:     What distinguishes this paper from the recent review articles by Li et al., 2021? “ZNF217: the cerberus who fails to guard the gateway to lethal malignancy”. Please include one or two sentences about it.

Response to comment 3 : To fulfill the reviewer’s request, we have modified the text lines 124-140, to include the information previously provided in our cover letter.

“Two main previous reviews have extensively depicted the role of ZNF217 in carcinogenesis by affecting the hallmarks of cancers (For review [1,2]), emphasizing ZNF217 importance as a prognostic biomarker for early prevention and as a therapeutic target. Recent literature provides, however, evidence of complex networks involving ZNF217 and epigenetic processes, as new key regulators of tumorigenesis. Increasing body of research indicates that epigenetics processes control ZNF217/Zfp217 expression levels and ZNF217/Zfp217-driven functions. Indeed, deregulated DNA methylation status at ZNF217 locus or complex cross-talk between ZNF217 and non-coding RNA networks could explain aberrant ZNF217 upregulation. On the other hand, ZNF217 could control gene expression signatures and molecular signaling for tumor progression and cell plasticity. ZNF217 has been shown to tune DNA methylation status at key gene promoters, to interfere with non-coding RNA or to refine the epitranscriptome.

Altogether, this topic having beeing poorly reviewed and appreciated, a better understanding and clear overview of the intricate interplay between ZNF217 and epigenetics networks in cancer is of utmost importance. The present review focuses on and discusses recent advances delineating the complex interaction between ZNF217 and epigenetic processes to orchestrate tumor progression.”

Comment 4:     It would be great if the authors included a figure that summarized all of the interlink signaling pathways in cancer through ZNF217, regardless of cancer type.

Response to comment 4 : To fulfill the reviewer’s request, we have now added in the revised version of the manuscript a new figure (numbered Figure 1 in the revised version of the manuscript), illustrating the signaling pathways known to be down-stream effectors of ZNF217 functions. All the remaining figures have been renumbered accordingly. The text has been modified accordingly (line 51)

Reviewer 3 Report

Submitted manuscript titled “The intricate interplay between the ZNF217 oncogene and epi-2 genetic processes shapes tumor progression”, by Pia Fahmé et al wrote a very detailed and qualitative review article on the epigenetic function of ZNF217. They described DNA methylation, epitranscriptome and non-coding RNA associated with ZNF217. It's a very good manuscript, but some recent research work is missing and needs to be added.

1.     Huaqing Ou et al. reported that Hsa_circ_0069094 showed an oncogenic effect in breast cancer by competitively binding miR-758-3p and activating the expression of ZNF217 (PMID: 36356557). This result should be added to the miRNA section.

2.     Waterbury et al. reported the interaction between ZNF217 and miR-130b-3p in a polycystic ovary syndrome model (PMID: 35668995). This result should be added to the miRNA section.

3.     An et al. reported that ZNF217 methylation status differs according to carcinoma subtype in the TCGA cohort (PMID: 35378249). Kidney renal clear cell carcinoma, kidney renal papillary cell carcinoma exhibited hypomethylation status in tumor cell samples compared to normal samples. This result should be added to the DNA methylation.

Author Response

Response to Reviewer 3’s comments

Reviewer 3 Comments and Suggestions for Authors

Submitted manuscript titled “The intricate interplay between the ZNF217 oncogene and epi-2 genetic processes shapes tumor progression”, by Pia Fahmé et al wrote a very detailed and qualitative review article on the epigenetic function of ZNF217. They described DNA methylation, epitranscriptome and non-coding RNA associated with ZNF217.

Comment 1 It's a very good manuscript,but some recent research work is missing and needs to be added.

  1. Huaqing Ou et al. reported that Hsa_circ_0069094 showed an oncogenic effect in breast cancer by competitively binding miR-758-3p and activating the expression of ZNF217 (PMID: 36356557). This result should be added to the miRNA section.
  2. Waterbury et al. reported the interaction between ZNF217 and miR-130b-3p in a polycystic ovary syndrome model (PMID: 35668995). This result should be added to the miRNA section.
  3. An et al. reported that ZNF217 methylation status differs according to carcinoma subtype in the TCGA cohort (PMID: 35378249). Kidney renal clear cell carcinoma, kidney renal papillary cell carcinoma exhibited hypomethylation status in tumor cell samples compared to normal samples. This result should be added to the DNA methylation.

Response to Comment 1:

First we would like to thank the reviewer for his/her positive comment.

Second, regarding the references suggested by the reviewer, please kindly find our response:

- Huaqing Ou et al. (PMID: 36356557) :

This paper has been very recently published while our manuscript was under the review process.

We are pleased to have now updated our manuscript with this reference in the text (lines 554-557) and in the revised Figure 4.

- Waterbury et al. (PMID: 35668995).

We respectfully believe that this reference is not appropriate for our review, as this paper describes mir-130b-3p, which targets DENND1A.V2 mRNA (and not ZNF217 mRNA). We have also the feeling that the apparent increase in miR-130b-3b expression levels upon ZNF217 ectopic expression has not been fully investigated and deciphered.

 We haven’t included this reference in our review and we thank the reviewer for his/her comment and understanding.

-An et al (PMID: 35378249).

We are very grateful to the reviewer for his/her comment, as we had indeed missed/skipped this reference. The reference and information are now provided in the revised version of our manuscript, line 171-175.

Reviewer 4 Report

This is a nice and timely review about the oncogenic properties of ZNF217 its role as a transcription factor and regulator of the epigenome and lncRNAs. The authors effectively summarize several studies of lncRNAs-miRNAs-mRNAs networks regulating ZNF217. I only have minor suggestions:

Lane 20: The sentence: the lack of identification of the full 3D-structure of ZNF217, makes it more beneficial to investigate mechanisms of regulation of ZNF217. Makes no sense to me.

Lane 189: what is positive proliferation control, why the word positive is needed in this sentence?

Lane 266: please defines aberrant deposition, more or less m6A deposition?

Lane 576: MCF-7 should be MCF-7

Author Response

Response to Reviewer 4’s comments

Reviewer 4 Comments and Suggestions for Authors

This is a nice and timely review about the oncogenic properties of ZNF217 its role as a transcription factor and regulator of the epigenome and lncRNAs. The authors effectively summarize several studies of lncRNAs-miRNAs-mRNAs networks regulating ZNF217. I only have minor suggestions:

Comment 1: Lane 20: The sentence: the lack of identification of the full 3D-structure of ZNF217, makes it more beneficial to investigate mechanisms of regulation of ZNF217. Makes no sense to me.

Response to comment 1: We have modified the sentence in the revised version of the manuscript, that is now: “Therapies targeting ZNF217 could be promising in cancer treatment, however, the lack of identification of the full 3D-structure of ZNF217 incites to investigate mechanisms of regulation of ZNF217 to delineate alternative candidate therapeutic strategies” (lines 19-22).

We hope to have clarified this sentence in response to the reviewer’s comment

Comment 2: Lane 189: what is positive proliferation control, why the word positive is needed in this sentence?

Response to comment 2: We agree with the reviewer and we have modified the sentence accordingly by deleting “positive” (line 206 in the revised version of the manuscript).

Comment 3: Lane 266: please defines aberrant deposition, more or less m6A deposition?

Response to comment 3: We thank the reviewer for his/her relevant comment. We have clarified the sentence that is now “Following melatonin/melatonin receptor 1 (MT1) interaction, low m6A deposition is observed, in particular to NANOG, SOX2, KLF4 and MYC mRNA, leading to increased mRNA stability [23].” (line 283 of the revised version of the manuscript).

Comment 4: Lane 576: MCF-7 should be MCF-7

Response to comment 4: The typo is now edited in the revised version of the manuscript (line 597).